# Uncertainty leads to persistent effects on reach representations in dorsal premotor cortex

Brian M Dekleva[1], Pavan Ramkumar[2], Paul A Wanda[3], Konrad P Kording[1,2,3,4], Lee E Miller[1,2,3]*

[1]Department of Biomedical Engineering, McCormick School of Engineering, Northwestern University, Evanston, United States; [2]Department of Physical Medicine and Rehabilitation, Feinberg School of Medicine, Northwestern University, Chicago, United States; [3]Department of Physiology, Feinberg School of Medicine, Northwestern University, Chicago, United States; [4]Department of Engineering Sciences and Applied Mathematics, Northwestern University, Evanston, United States

**Abstract** Every movement we make represents one of many possible actions. In reaching tasks with multiple targets, dorsal premotor cortex (PMd) appears to represent all possible actions simultaneously. However, in many situations we are not presented with explicit choices. Instead, we must estimate the best action based on noisy information and execute it while still uncertain of our choice. Here we asked how both primary motor cortex (M1) and PMd represented reach direction during a task in which a monkey made reaches based on noisy, uncertain target information. We found that with increased uncertainty, neurons in PMd actually enhanced their representation of unlikely movements throughout both planning and execution. The magnitude of this effect was highly variable across sessions, and was correlated with a measure of the monkeys' behavioral uncertainty. These effects were not present in M1. Our findings suggest that PMd represents and maintains a full distribution of potentially correct actions.

*For correspondence: lm@northwestern.edu

**Competing interests:** The authors declare that no competing interests exist.

## Introduction

Each motor action we perform reflects only one of the many available or considered actions. In some situations, the full set of potential actions comprises a set of discrete choices (e.g., which of these three apples should I pick?). In these cases, the task for the sensorimotor system is to evaluate each option and decide which will lead to the most favorable outcome. However, these 'target selection' situations represent only one type of motor related decision-making. In many other scenarios the sensorimotor system cannot simply *select* between multiple explicit options, but instead must *estimate* the best action based on continuous – and often noisy – sensory information and learned experience. Reaching toward a familiar object seen only in the peripheral vision, or under poor illumination is one such example.

Though target selection represents only one type of sensorimotor task, it dominates the current literature on neural correlates of motor-related decision making. This is true for both eye movements (*Basso and Wurtz, 1997*; *Britten et al., 1996*; *Fetsch et al., 2011*; *Newsome and Britten, 1989*; *Shadlen and Newsome, 2001*) and reaching (*Bastian et al., 2003*; *Cisek and Kalaska, 2005*; *Coallier et al., 2015*; *Messier and Kalaska, 2000*; *Thura and Cisek, 2014*). These studies vary significantly in the methods by which they provide cues to elicit a motor response. The cues may indicate different parameters of the action, such as the direction or extent of the movement

**eLife digest** Whether it is trying to find the light switch in a dimly lit room or reaching for your glasses when you wake in the morning, we often need to reach toward objects that we cannot see clearly. In these situations, we plan our movements based both on the limited sensory information that is available, as well as what we have learned from similar situations in the past.

The brain areas involved in using information to decide on the best movement plan appear to be different from those involved in actually executing that plan. One area in particular, called the dorsal premotor cortex (or PMd), is thought to help a person decide where to reach when they are presented with two or more alternative targets. However, it was not known how this brain area is involved in choosing a direction to reach when the targets are fuzzy, or unable to be seen clearly.

Dekleva et al. trained Rhesus macaque monkeys to reach in various directions, towards targets that were represented by fuzzy, uncertain visual cues. These targets were not simply positioned randomly; instead they were more likely to require reaches in certain directions over other directions. Because there were many such training and experimental sessions, the monkeys were able to learn where targets were more likely to be located. Dekleva et al. found that, like humans, the monkeys combined this knowledge from previous experience with the fuzzy visual information; like people, the monkeys also weighted each source of information based on how well they trusted it. For example, blurrier targets were treated as less trustworthy.

Further analysis showed that neurons in the PMd signaled the chosen direction well before the monkey began to reach. However, throughout the entire time the monkey was reaching, the same neurons also seemed to hold in reserve the other, less likely reach directions. In contrast, neurons in the area of the brain that directly controls movement – the primary motor cortex – only ever signaled the direction in which the monkey actually reached.

Further work is now needed to understand the decision-making process that appears to start in the PMd and resolve in the primary motor cortex. In particular, future experiments could explore why the retained information about other possible reach decisions persists throughout the movement, including if this helps the individual to rapidly correct errors or to slowly improve movements over time.

(*Bastian et al., 2003*; *Crammond and Kalaska, 1994*; *Gail et al., 2009*; *Messier and Kalaska, 2000*; *Welsh and Elliott, 2005*). They can be discrete (*Meegan and Tipper, 1998*; *Thura and Cisek, 2014*; *Wood et al., 2011*) or continuous (*Gold and Shadlen, 2001*; *Hernández et al., 2010*; *Resulaj et al., 2009*), and can even span different sensory modalities (*Hernández et al., 2010*; *Romo et al., 2004*). However, all share a common characteristic: the action is directed towards one of several mutually exclusive targets. This mutual exclusivity is a constraint specific to the task of target selection and does not exist in target estimation, since no explicit options are presented. It is therefore not obvious how the results from target selection tasks may or may not extend to the case of target estimation.

In both target selection and estimation, there is some degree of uncertainty in the decision making process as well as in the final decision itself. This uncertainty largely depends on the ambiguity of the available cues. If the task includes a completely unambiguous cue indicating the correct choice, the decision will contain practically no uncertainty whatsoever. For example, one standard multiple-target selection task used in non-human primate reaching studies (e.g., *Bastian et al., 2003*; *Cisek and Kalaska, 2005*) briefly presents a monkey with two or more potential reach targets before indicating the correct one. In this situation the animal may be initially uncertain about which target is correct, but that uncertainty vanishes with the disambiguating cue. Variants of this task provide more ambiguous cues and allow the animal to choose one of two targets while still unsure about the correct choice (*Coallier et al., 2015*; *Thura and Cisek, 2014*), which results in decisions that are made despite a lingering uncertainty.

Studies of reach-related brain areas during target selection tasks have suggested that the dorsal premotor cortex (PMd) plays a significant role in sensorimotor decision-making. Historically, PMd has been viewed as a movement planning area, displaying activity consistent with a representation

of upcoming movements to visual targets (*Cisek et al., 2003*; *Shen and Alexander, 1997*; *Weinrich and Wise, 1982*). Later studies showed that these pre-movement representations can include multiple simultaneous potential targets (*Cisek and Kalaska, 2005*) and reflect motor plans even in the absence of visual targets (*Klaes et al., 2011*). Furthermore, the representations during multiple-target tasks are modulated by decision-related variables (*Coallier et al., 2015*; *Pastor-Bernier and Cisek, 2011*). These more recent results are consistent with an interpretation that activity in PMd modulates with the complexity (or uncertainty) of a motor decision.

In general, sensorimotor decision-making should take into account the uncertainty present in all task-relevant information sources – namely the current sensation and prior experience. When sensation provides a highly reliable action cue (e.g., when reaching toward a well-lit, foveated object), it can be used exclusively to plan and execute the appropriate motor output. However, as uncertainty in sensation increases, it becomes more beneficial to combine sensory information with information learned through prior experience. The optimal method for integrating sensory and prior information was formulated centuries ago as Bayes' theorem (*Bayes and Price, 1763*). A direct application of Bayes' theorem states that cues should be weighted in inverse proportion to their variance (*Knill and Saunders, 2003*; *Körding and Wolpert, 2006*). The Bayes optimal decision will lead to better results than either cue alone, but will still contain a degree of uncertainty.

Bayesian models have been used to describe human behavior in a wide array of psychophysical studies, including visual (*Knill and Saunders, 2003*; *Mamassian and Landy, 2001*; *Weiss et al., 2002*), auditory (*Battaglia et al., 2003*), somatosensory (*Goldreich, 2007*), cross-modal (*Alais and Burr, 2004*; *Ernst and Banks, 2002*; *Gu et al., 2008*; *Rowland et al., 2007*), and sensorimotor (*Greenwald and Knill, 2009*; *Körding and Wolpert, 2004*; *Trommershäuser et al., 2008*; *van Beers et al., 2002*) applications. In these tasks, behavior generally matched the predictions of various Bayesian models of optimal performance, which has been taken as evidence that the brain does indeed incorporate information about the relative uncertainty of various cues when planning and executing movements.

To probe the effect of target estimation uncertainty on M1 and PMd, we designed a task in which monkeys estimated the location of reach targets using knowledge of the average target location (learned through experience) and noisy visual cues. Although M1 activity appeared to reflect only the direction of the executed reach, we found that the monkeys' uncertainty about where to reach correlated with changes in PMd activity during both movement planning and execution. The magnitude of these uncertainty-related effects in PMd was spatially tuned. Neurons whose strongest response direction (their preferred direction, or PD) was aligned with the planned reach direction remained largely unchanged, while neurons with PDs opposite the reach direction experienced a significant increase in activity with increased uncertainty. Neurons with intermediate PDs displayed somewhat smaller uncertainty-related effects. The uncertainty-related change in this off-direction neural activity varied considerably across sessions, not only because of experimentally altered prior and likelihood uncertainty, but also apparently because of the monkeys' own subjective uncertainty in their final action decisions. We found that the magnitude of these cross-session activity differences correlated with estimates of the monkey's decision-related uncertainty.

## Results

### Task performance during reaching to certain and uncertain targets

Our goal in this study was to understand the effect of uncertainty on arm movement representations in the motor system. To this end, we designed a behavioral task in which monkeys (one rhesus macaque, one cynomolgus macaque) made decisions about where to reach using a planar robotic manipulandum, based on the learned history of target distributions and uncertain visual cues. During the first block of trials, the monkeys made center-out reaches with an instructed delay to well-specified (zero uncertainty) targets that were randomly distributed across eight locations (*Figure 1A*, top). In the second block of trials, the target locations were randomly drawn from a circular normal (von Mises) *prior* distribution centered on a single direction that remained constant for the remainder of the session. Additionally, the monkey did not receive veridical feedback about the location of the target, but instead saw a noisy distribution of five (monkey M) or ten (monkey T) lines (*Figure 1A*, bottom). These lines were drawn from a *likelihood* distribution – also von Mises –

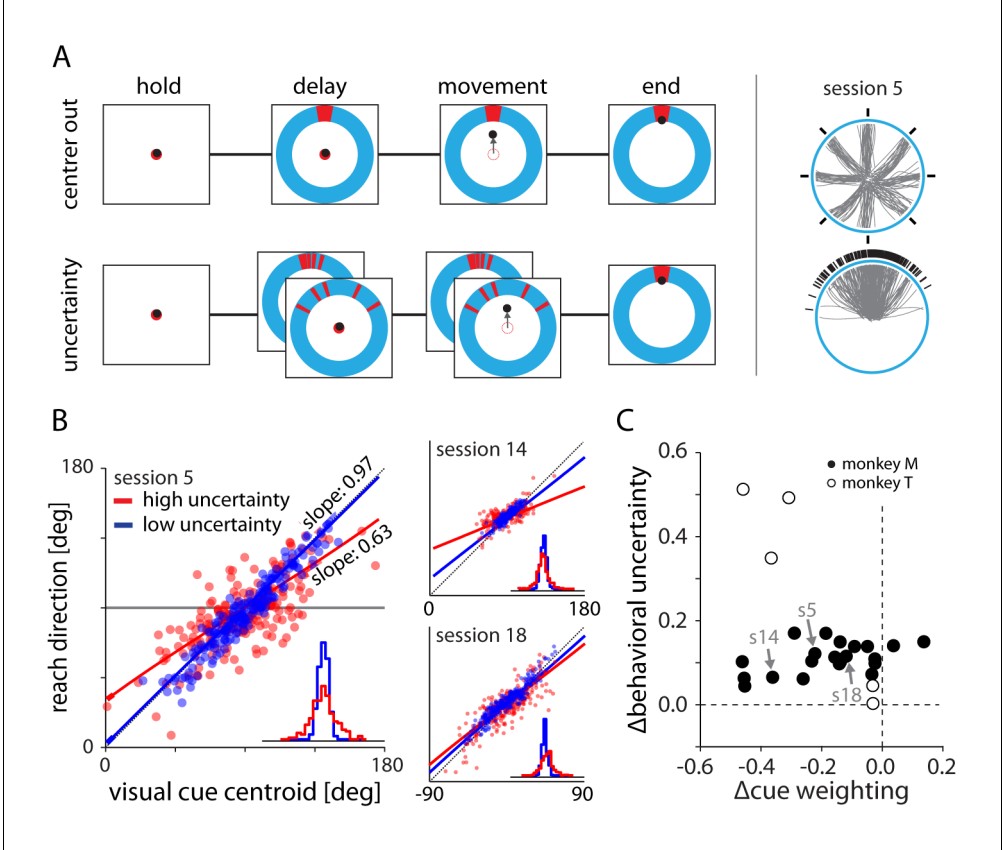

**Figure 1.** Experimental setup and behavior. (**A**) Monkeys made planar center-out reaches with instructed delay to visual targets. Illustrations on right show target locations (black) and reach trajectories (gray) for trials in the center out and uncertainty blocks for an example session. In the center-out block, targets were distributed uniformly across eight directions and were cued with no uncertainty. In the uncertainty block, targets were sampled from a von Mises distribution and cued with stochastically sampled lines with either low or high variance. (**B**) Scatter plots of cue centroid versus reach direction for three sessions, with each dot representing a single trial. Under high uncertainty, the endpoints reflected an increased bias toward the average target location – indicated by a reduction in slope – and increased variability surrounding the fit line. (**C**) With the exception of two datasets from monkey M, fits to the behavioral scatter plots reveal reduced slope (negative Δcue weighting) for higher uncertainty targets. All datasets show greater residual variance with greater uncertainty.

The following source data is available for figure 1:

**Source data 1.** Experimental details for all sessions.

centered on the correct target location, providing the monkey with noisy information about the target location. Each session contained at least two likelihood distributions of low and high variance, randomly interleaved across trials.

Therefore, during uncertainty trials, the monkey had two pieces of information available to estimate the target location: (1) the noisy visual cue and (2) a learned estimate of the distribution of previous target locations. According to Bayes' rule, optimal performance on the task would require the monkey to use the centroid of the displayed line segments (its likelihood estimate) and the average target location (prior estimate), weighted according to the inverse of their variances. In general, this means that using an appropriately weighted sum of both the likelihood and prior estimates will, on average, result in smaller errors than either cue alone.

Fits to the scatter plot between the centroid of the visual cue and the reach direction reveal the monkey's relative weighting of the visual cue (the likelihood) and its estimate of the average target location (the prior; see Materials and methods for more information). A fitted line with a slope of zero would indicate complete reliance on the prior, while a slope of one would indicate reliance only

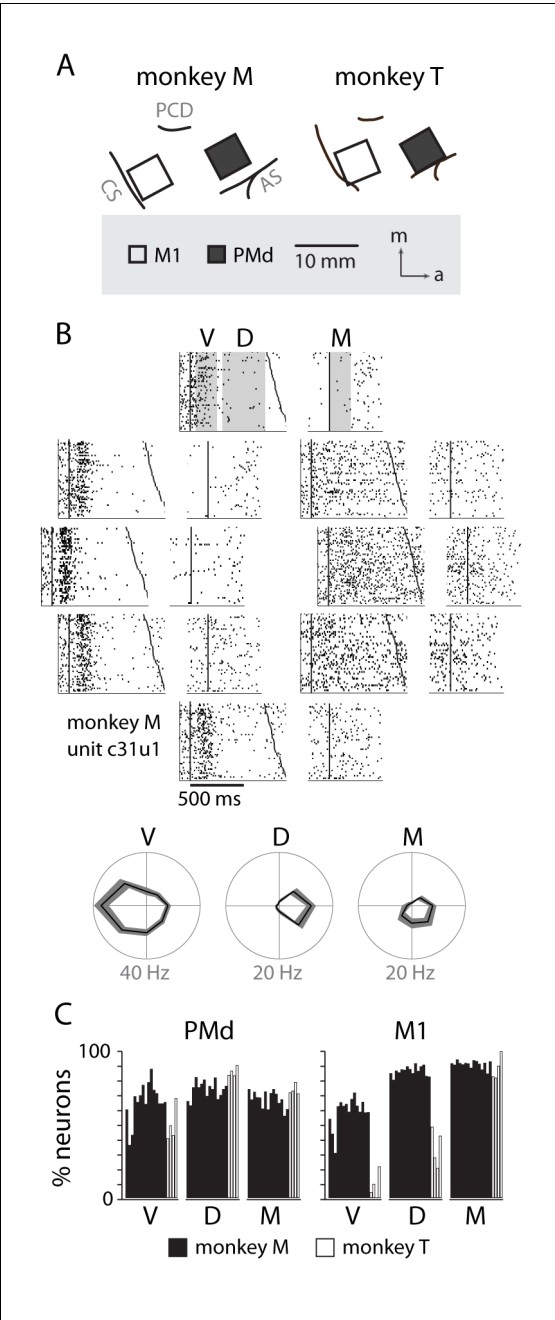

**Figure 2.** Neural recordings and directional tuning. (**A**) Each monkey was implanted with two 96-channel microelectrode arrays, targeting the primary motor cortex (M1) and dorsal premotor cortex (PMd). (**B**) An example raster of a neuron in PMd displaying directional tuning, summarized below in three temporal periods: visual (V), delay (D) and movement (M). (**C**) Percentage of neurons from each session with significant tuning in each of the temporal periods.

on the likelihood. Panel B of *Figure 1* shows several representative sessions. In each, the monkey relied more on the visual cue when its uncertainty was low (blue symbols) than when it was high (red symbols). We summarized the difference in visual cue weighting between the uncertainty conditions (Δcue weighting) for each session by subtracting the slopes of the fitted lines. The negative values of Δcue weighting in *Figure 1C* reveal that both monkeys almost always relied less on the visual cue during high uncertainty trials. This indicates that the monkeys combined information from both the

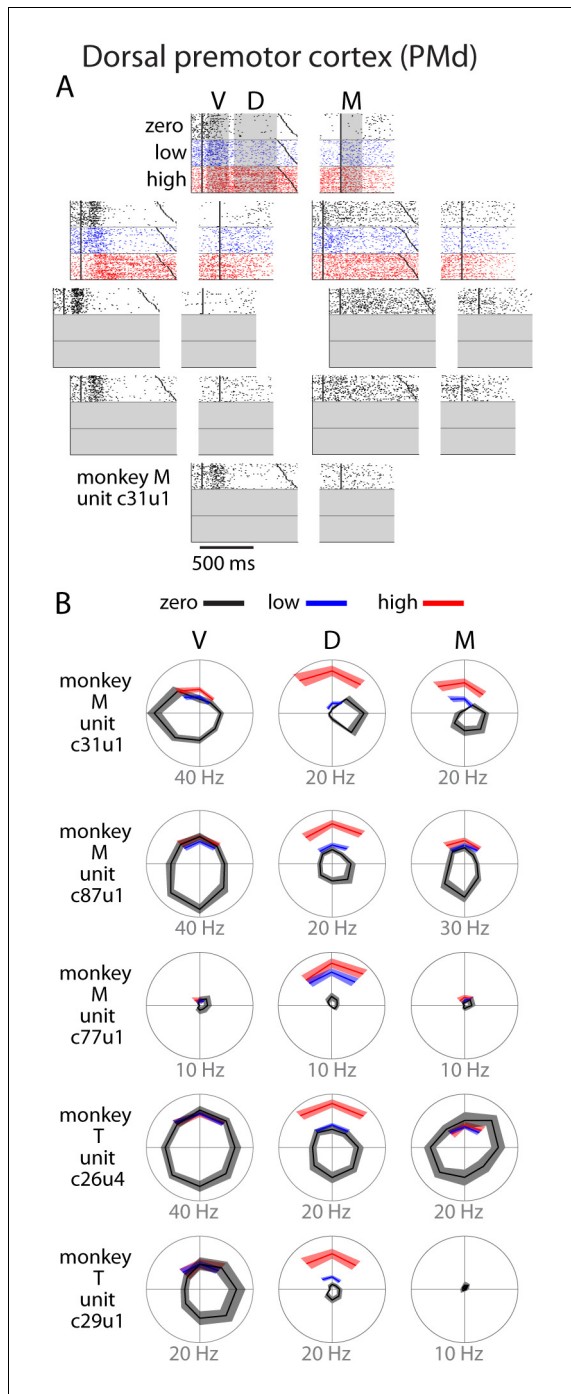

**Figure 3.** Single unit activity in PMd. (**A**) Raster plot for an example neuron. Activity is aligned to either the visual cue appearance (left) or movement onset (right). Colors indicate zero (black), low (blue), and high (red) uncertainty conditions. Dark black points indicate target onset, go cue, and movement onset (**B**) Directional tuning for other example neurons. Due to the nature of the task, reaches made during uncertain conditions with a non-uniform prior did not span all directions. Many neurons showed an increase in delay (D) or movement (M) activity as a function of uncertainty. Bounds on the tuning plots represent bootstrapped 95% confidence of the mean estimate.

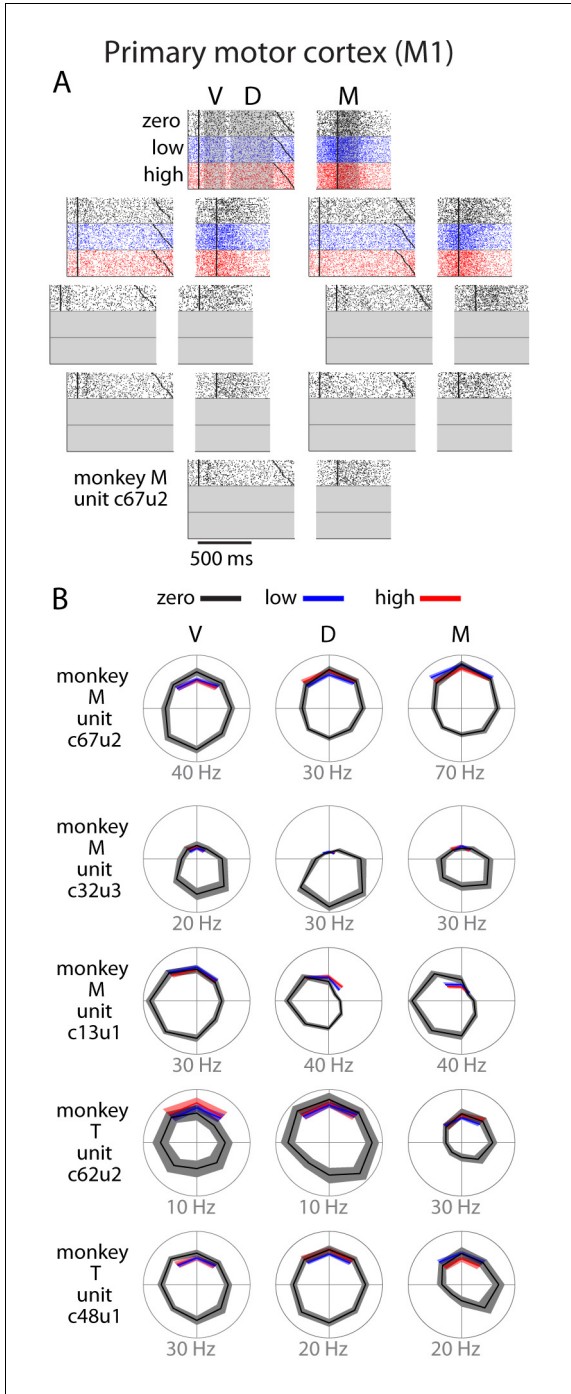

**Figure 4.** Single unit activity in M1. (**A**) Raster plot for an example neuron with same conventions as *Figure 3*. (**B**) Directional tuning for other example neurons. In general, M1 activity was well-modulated by reach direction, but appeared to be largely unaffected by the uncertainty condition. Bounds on the tuning plots represent bootstrapped 95% confidence of the mean estimate.

displayed lines and the average target location in a Bayesian-like manner to estimate the location of the required reach target.

Although there was a general tendency towards lesser weighting of the visual cue when it was more uncertain, there was a great deal of variability in that trend across sessions. In some instances, fits to the two uncertainty conditions revealed large differences in visual cue weighting (*Figure 1B*

red and blue fitted slopes, session 14) while in others the relative weighting was nearly identical (*Figure 1B*, session 18). Similarly, the uncertainty in the final estimate (as measured using the variance of the fit residuals) was sometimes very different between two conditions (*Figure 1* inset distributions, session 5) and sometimes nearly identical (*Figure 1*, session 14). We characterized the total difference in this behavioral uncertainty between the two conditions (Δbehavioral uncertainty) for each session by subtracting the angular dispersion of the residuals. These two within-session metrics (Δcue weighting and Δbehavioral uncertainty) were very weakly correlated for monkey M and negatively correlated for monkey T (*Figure 1C*). This variability provided a diverse set of uncertainty-related behavioral effects on which to examine neural activity.

## Neural activity

During the center out block of trials (zero uncertainty, eight discrete targets) many neurons in PMd displayed a robust burst of activity directly following presentation of the visual cue, followed by a more moderate, tonic response for the remainder of the delay period (e.g., *Figure 2B*). We more formally described the population trends by calculating the percentage of neurons tuned in the visual (V), delay (D), and movement (M) time periods. The results for each session are shown in *Figure 2C*. We performed the same analysis for M1 neurons (*Figure 2C*, right). In general, M1 displayed a bias toward delay and movement period tuning while PMd showed about equal percentages of tuned neurons for each time period.

During the remaining experimental blocks consisting of uncertain targets, we found many neurons in PMd to be more active during high uncertainty trials than low uncertainty trials (red vs. blue in *Figure 3*). This effect was most prominent during the delay (D) period, with some carryover into movement (M). Some neurons that had been essentially inactive during the block of zero-uncertainty reaches became strongly active during the delay period of high-uncertainty trials (e.g., c77u1 and c29u1, *Figure 3*). We also noted that there was a greater tendency for increased activity in neurons with PDs not aligned to the direction of movement (e.g., c31u1 and c87u1, *Figure 3*). Importantly, we found that greater uncertainty only ever led to increased activity.

M1 neurons did not display nearly the same degree of modulation with uncertainty as PMd neurons (*Figure 4*). We observed neurons with strong directional tuning in all time periods, but this tuning was consistent across all uncertainty conditions. In general, analysis of single unit behavior suggested that M1 activity reflected only the reach direction and was largely unaffected by uncertainty.

## Quantifying effects of uncertainty on firing rates

The anecdotal observations in *Figures 3* and *4* strongly suggest that higher uncertainty leads to increased neural discharge in PMd but not in M1. Additionally, the magnitude of the uncertainty-related effect in individual PMd neurons was dependent on the neurons' tuning characteristics. A neuron experienced the greatest uncertainty-related activity increase when the reaches were directed away from its preferred direction. To further examine this relationship between tuning and uncertainty-related activity changes, we created spatiotemporal activity maps for both cortical areas in the manner of *Cisek and Kalaska (2005)* (*Figure 5*). We binned each neuron's responses based on the angle between its PD and the reach direction. We then averaged across trials, resulting in population activity profiles centered on reach direction.

In the zero-uncertainty condition, many PMd neurons displayed a burst of activity directly following cue appearance. This quickly resolved into a clear, maintained representation of the upcoming reach direction throughout the remainder of the delay and movement periods (*Figure 5A*, top left). In contrast, M1 activity built more slowly as the trial evolved, ultimately producing a strong spatial representation of the executed reach direction (*Figure 5A*, top right). However, while the recruitment of M1 neurons during low and high uncertainty conditions was similar (*Figure 5A*, right), the representation in PMd differed significantly across these conditions. During high uncertainty trials, the representation of the reach direction in the delay period was present but significantly less distinct, most notably due to increased activity in neurons with PDs far away from the reach direction (*Figure 5A*, bottom left). We partitioned the neurons into three groups for each trial: same direction (SD; preferred direction within 45 degrees of the reach direction), opposite direction (OD; preferred direction within 45 degrees of the anti-reach direction), and orthogonal direction (ORTH; preferred

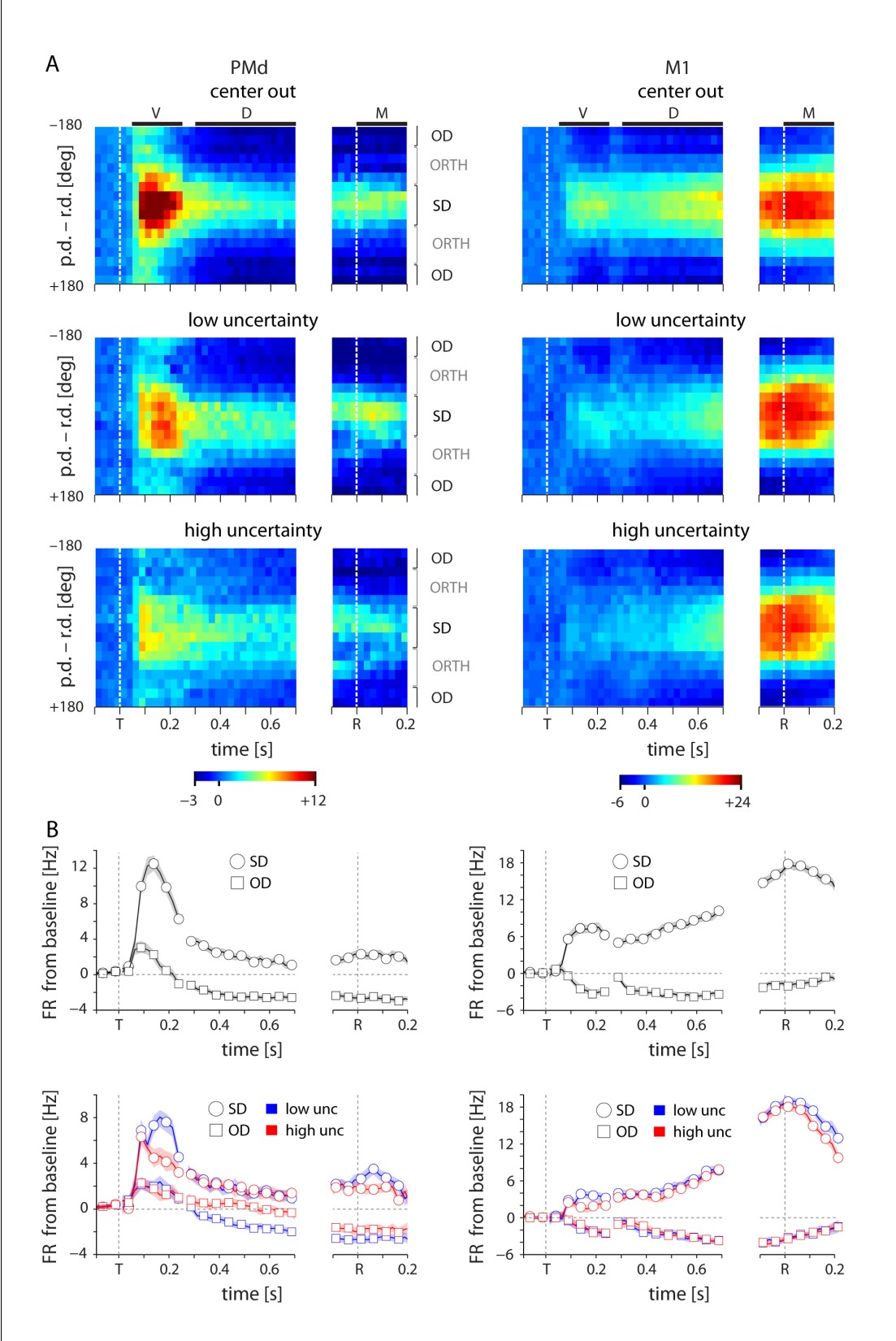

**Figure 5.** Tuning-related changes in activity with uncertainty. (**A**) Spatiotemporal activity maps for PMd and M1. Neurons were binned on each trial by the distance between their preferred directions and the reach direction. Color indicates average change in firing rate from baseline in spikes per

*Figure 5 continued on next page*

*Figure 5 continued*

second. Left and right plots in each panel are aligned to target onset (T) and reach onset (R) respectively. (**B**) Average change from baseline for SD and OD neurons in the initial center-out block (zero uncertainty; top) and subsequent blocks with low (bue) and high (red) uncertainty targets (bottom). High uncertainty trials resulted in reduced early activity for both SD and OD neurons in PMd, but an increase in OD activity for the remainder of the delay and movement phases. ORTH neurons were omitted for visibility. Error bars represent bootstrapped 95% confidence bounds on the mean estimate. For all plots, PDs were calculated separately for visual, delay, and movement epochs.

direction within 45 degrees orthogonal the reach direction). After averaging the activity of these populations, it became clear that while both SD and OD neurons in PMd were less active immediately after target appearance in high uncertainty trials, the OD neurons showed higher activity in the subsequent D and M periods. Thus the main delay-period effect of higher target uncertainty was an increase in the PMd activity in neurons with preferred directions away from the reach direction.

To summarize this uncertainty effect over sessions, we calculated the difference in average firing rates between low and high uncertainty conditions for SD, ORTH, and OD neurons. In most sessions, ORTH and OD activity during the delay and movement periods was significantly greater in the high uncertainty condition, while SD activity showed little change (*Figure 6A* – monkey M; *Figure 7A* – monkey T). However, the increase in OD activity varied considerably across sessions. We reasoned that the sessions with the greatest OD activity differences might correspond to the sessions with the greatest differences in the monkeys' uncertainty. To test this, we calculated the difference in behavioral uncertainty (Δbehavioral uncertainty) between uncertainty conditions for each session (see Materials and methods: *behavioral task*). By plotting the activity differences as a function of Δbehavioral uncertainty, we found strong positive correlations for OD activity, but none for SD (*Figure 6B* – monkey M; *Figure 7B* – monkey T). For monkey M, the slope of the relation increased from SD to ORTH to OD neurons (*Figure 6B*), consistent with the single-session example shown in *Figure 5*. We found very similar effects of uncertainty among OD neurons for monkey T (*Figure 7B*). These findings suggest that as the monkeys became less certain about their decision of where to reach, the representations of less likely reach directions increased.

We also found that the tuning-related effect of uncertainty persisted throughout the entirety of movement planning and even after the initiation of the reach. We applied the analysis in *Figure 6B* to different time periods throughout the trial and plotted the slopes (*Figure 6C*) and $R^2$ (*Figure 6D*) relating Δbehavioral uncertainty to changes in SD, ORTH, and OD activity. For both monkeys, the difference in OD activity first displayed a significant correlation with Δbehavioral uncertainty during the visual period (*Figures 6,7*, panels C and D). This effect persisted throughout the remainder of the delay period and the initiation of movement. ORTH activity displayed a similar trend but with a consistently shallower slope, indicating a weaker effect of uncertainty. SD neurons never displayed any significant correlation with uncertainty. For monkey T, only OD activity was consistently correlated with uncertainty throughout the delay and movement periods (*Figure 7C,D*). Thus it appears that movement representations in PMd remain affected by decision-related uncertainty leading up to and throughout execution of a movement.

There was also substantial cross-session variability in the M1 firing rates between high and low uncertainty. For monkey M, SD activity was generally lower for high uncertainty trials and OD activity was slightly higher (*Figure 8A*). However, there was rarely any correlation between the firing rate difference and the difference in behavioral uncertainty. For monkey M, SD activity was negatively correlated with uncertainty at the beginning of the delay period (300–400 ms following target appearance; *Figure 8C*). This effect dissipated quickly and was never observed for monkey T. As a result, we conclude that behavioral uncertainty had no significant effect on M1 activity during movement planning or execution.

Although the correlations between behavioral uncertainty and OD activity in PMd were significant, we considered the possibility that the neural effects were actually driven by the monkeys' relative weighting of the visual and prior information. To disassociate these two possibilities, we examined the independent correlations of OD activity with each of the two metrics in selectively subsampled groups of sessions. When we chose sessions that caused Δbehavioral uncertainty and Δcue weighting to be highly correlated (further exaggerating their normal relation), both metrics explained the change in OD activity (*Figure 9A*). However, for subsampled groups of sessions with

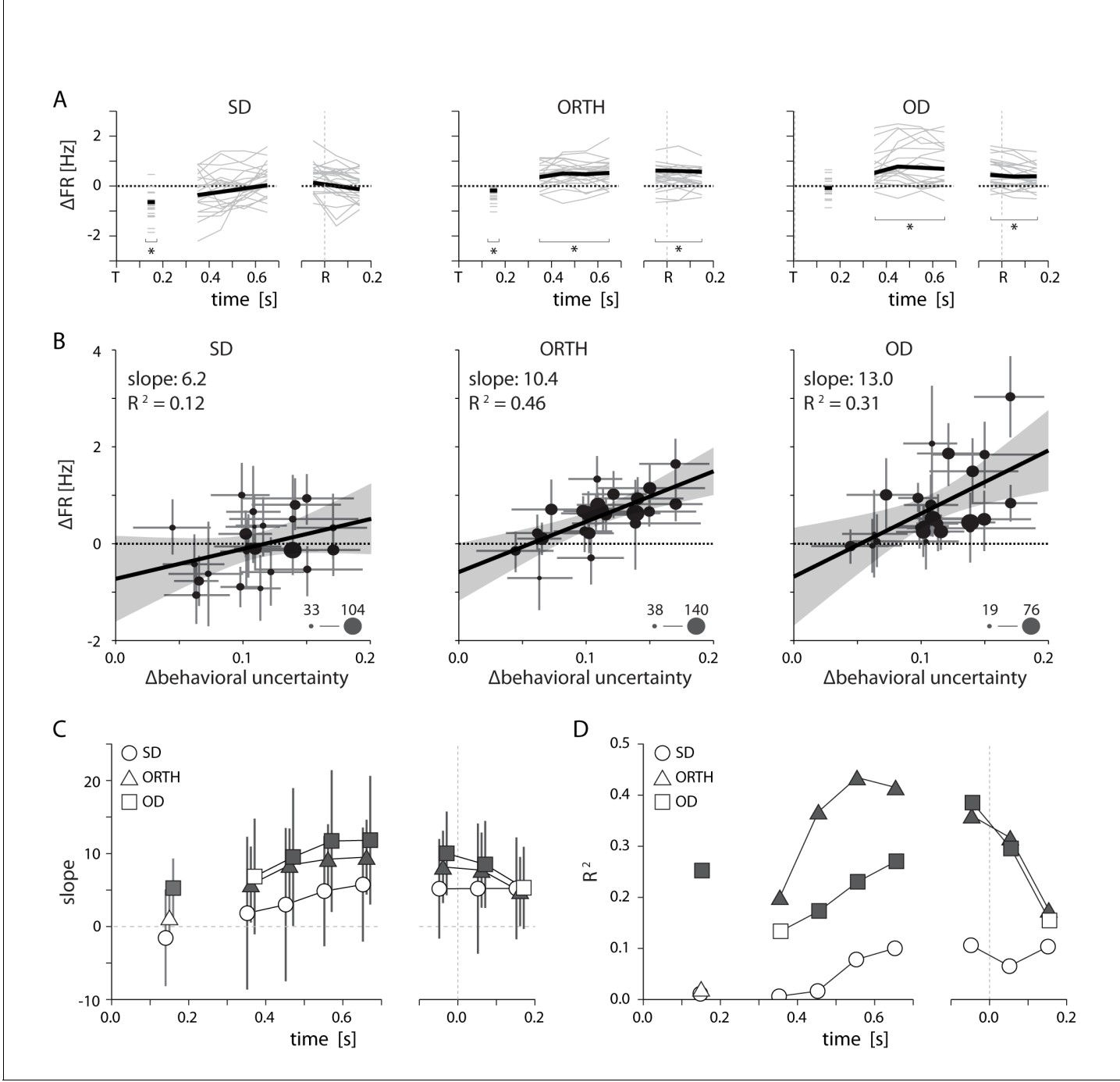

**Figure 6.** Relationship between PMd activity and behavioral uncertainty. (A) Thin lines indicate the average difference in firing rate between high and low uncertainty trials for individal sessions. Heavy lines mark the mean across sessions. While SD neurons displayed an average change near zero, activity for ORTH and OD neurons was consistently higher for high uncertainty trials (B) Differences in firing rate between high and low uncertainty conditions as a function of the difference in behavioral uncertainty for a single time window 500–700 ms after target appearance. The correlation was weak for same-direction neurons, but strongly positive for orthogonal- and opposite-direction neurons. Thus, the greater the difference in behavioral uncertainty, the larger the difference in activity for ORTH and OD neurons. Marker size indicates the number of contributing neurons for each session (C) The slopes from *B* calculated during the visual period (50–250 ms after target appearance; left) and for 100 ms time windows throughout the delay (middle) and movement (right) periods. The larger effect of behavioral uncertainty on OD and ORTH activity compared to SD activity persisted throughout planning and execution. (D) $R^2$ values for the linear fits in C. Filled symbols in **C** and **D** represent significant correlations, $p<0.05$. All error bars represent bootstrapped 95% confidence bounds on the mean estimates.

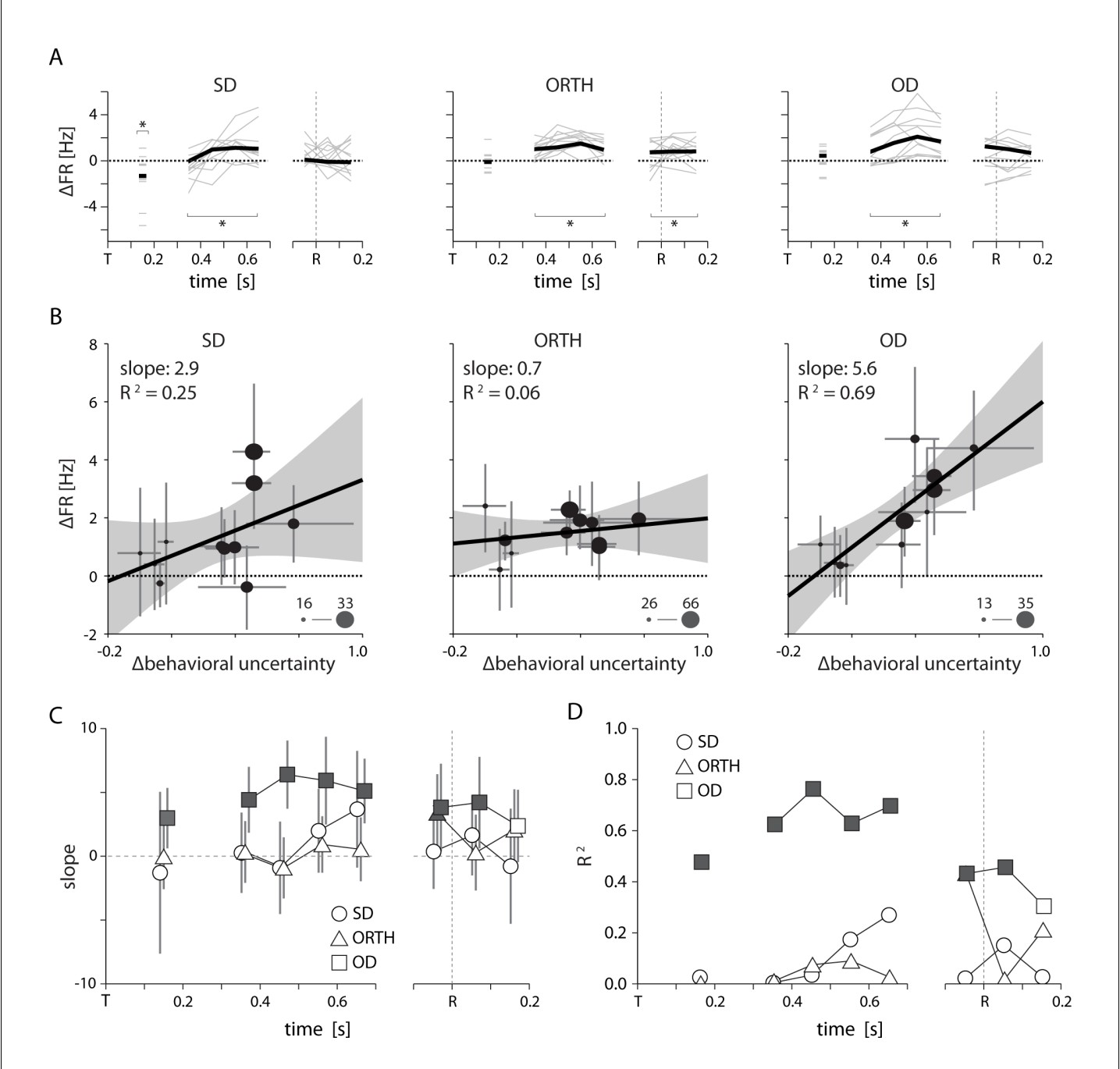

**Figure 7.** Summary of uncertainty related effects in PMd for Monkey T. All conventions as in *Figure 6*. Although we had only five sessions for monkey T, by splitting larger sessions into multiple blocks we obained 11 total data points. Specifics are given in *Figure 7—source data 1*.

The following source data is available for figure 7:

**Source data 1.** Subsampling of sessions for monkey T.

poor correlation between the two metrics, only Δbehavioral uncertainty remained well correlated with OD activity (*Figure 9B*). In fact, differences in OD activity correlated better with Δbehavioral uncertainty than with Δcue weighting for almost any randomly subsampled group of sessions for either monkey (*Figure 9C,D*). This suggests that the firing rate changes in PMd actually reflect

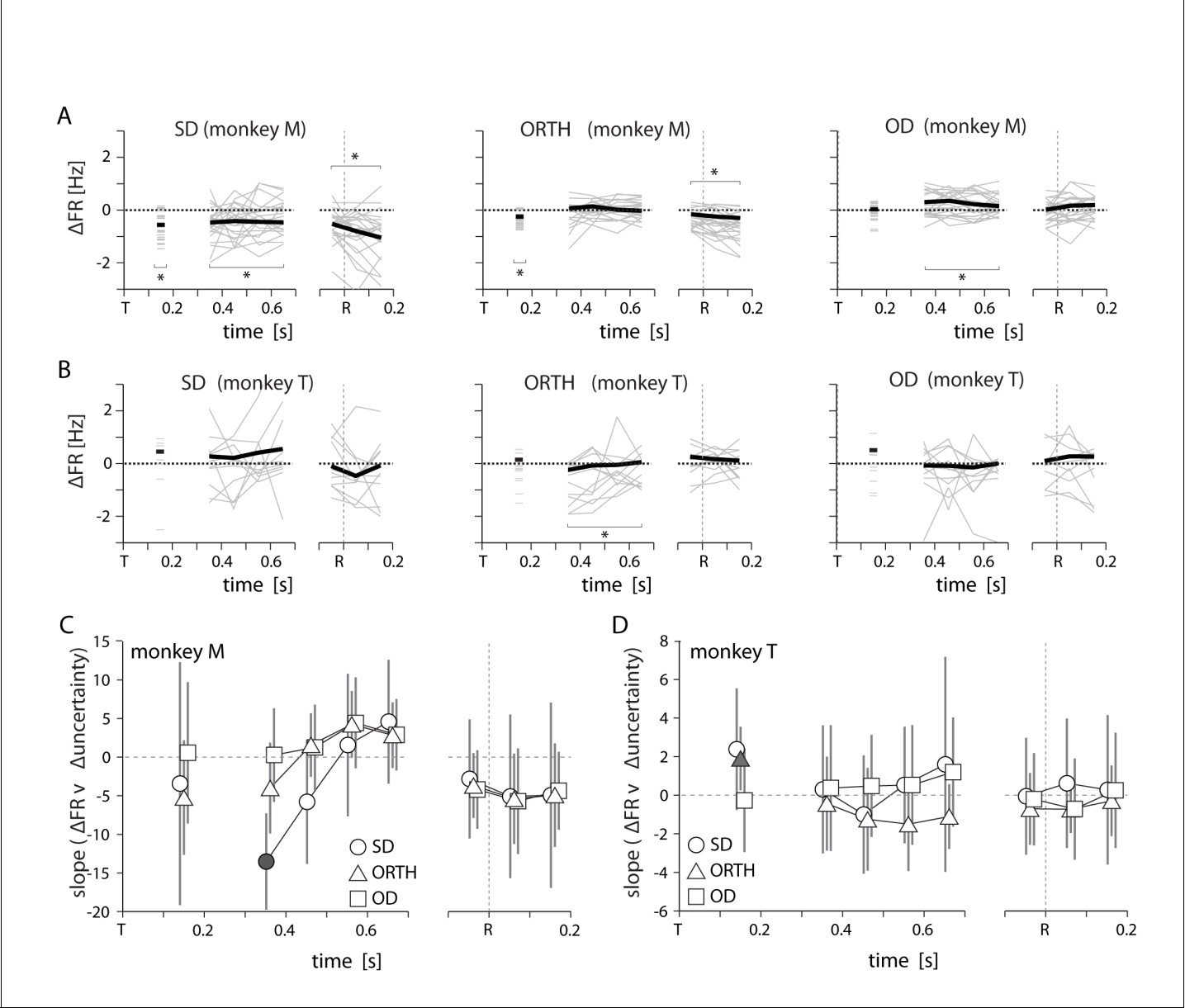

**Figure 8.** Summary of uncertainty-related actvity in M1 for both monkeys. All conventions as in *Figures 6*,*7*. Specifics of how we obtained datapoints for monkey T in panels B and D are given in *Figure 7—source data 1*.

differences in the monkeys' uncertainty about their decisions, rather than the weights applied to either visual or prior information leading to those decisions.

Another way of examining the evolution of target-related information in M1 and PMd is to use the neural activity to predict the monkey's choice of reach direction. For a representative session, we found that although it was possible to predict the monkey's reach direction from PMd activity, the predictions were consistently less accurate for high uncertainty trials than for low uncertainty trials (*Figure 10A*, left). Accuracy rather rapidly reached these levels within about 200 ms of target appearance, but then increased more slowly throughout the remainder of the trial. On the other hand, the ability to decode reach direction from M1 improved steadily through the delay period (*Figure 10A*, right). This was true for both high and low uncertainty trials, with only slightly higher delay-period decoding accuracy for low uncertainty trials. At the time of movement initiation, the M1 decoder was equally accurate for both conditions.

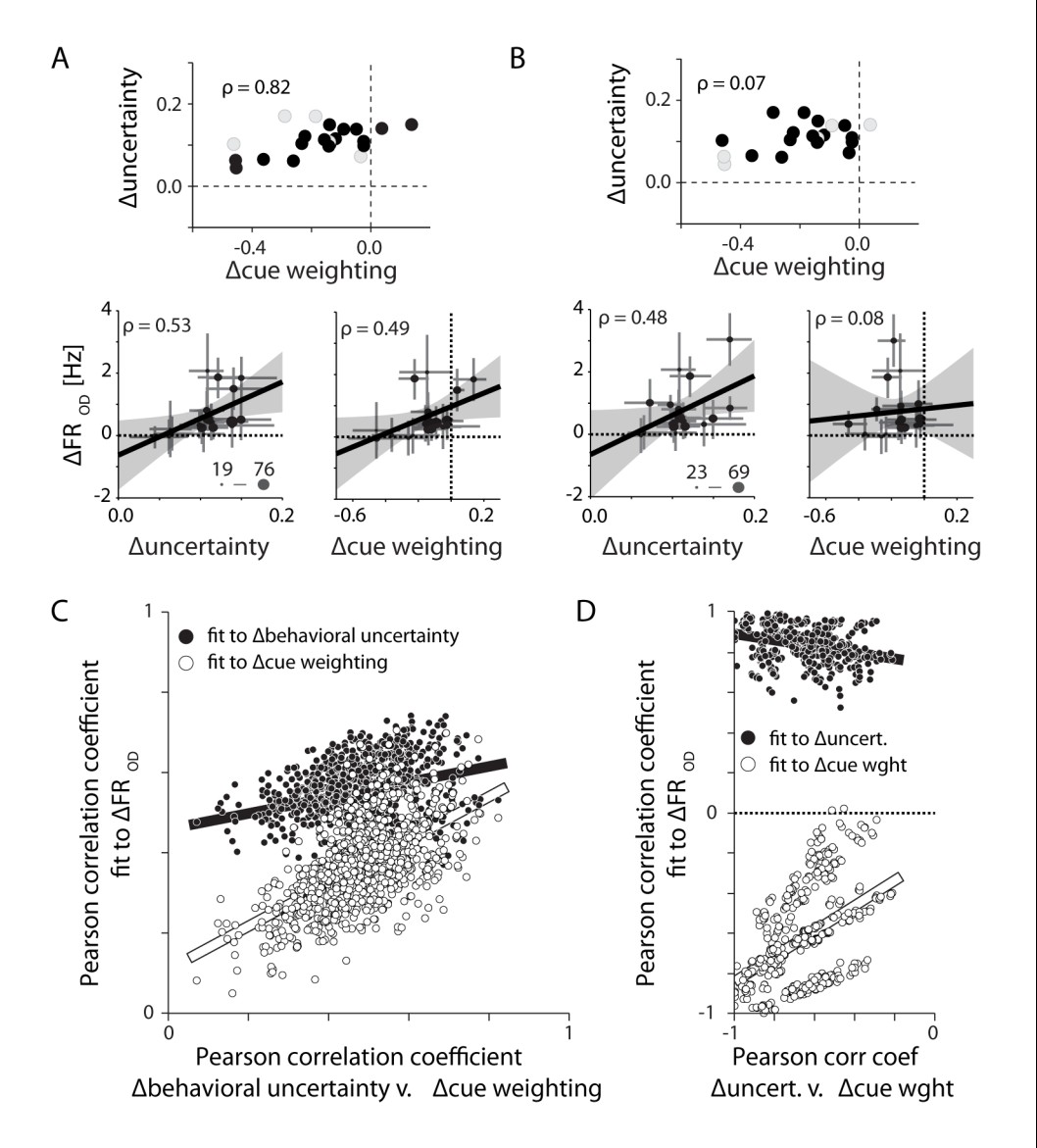

**Figure 9.** Differences in PMd activity correlate with differences in behavioral uncertainty rather than differences in the weighting of the visual cue. (**A**) Eighteen sessions (filled symbols) selected for monkey M in order to increase the correlation between Δbehavioral uncertainty and Δcue weighting (top). Across these select sessions both metrics could explain the observed differences in OD activity (bottom). (**B**) Alternate subsampling that *minimized* the correlation between the two behavioral metrics (top). This resampling did not change the correlation between changes in OD activity and Δbehavioral uncertainty (lower left). However, it eliminated the correlation between Δcue weighting and OD activity (lower right). (**C**) Correlations of OD differences with Δbehavioral uncertainty (filled) and Δcue weighting (open) for 1000 unique 18-session subsamples. Each is plotted against the correlation between Δbehavioral uncertainty and Δcue weighting. The correlation with Δbehavioral uncertainty was consistently stronger than with Δcue weighting. The correlation with Δcue weighting was only strong when Δcue weighting and Δbehavioral uncertainty were well correlated with each other. (**D**) Same as in **C**, but for monkey T. Each subsample contains six trial blocks. Unlike monkey M, Δcue weighting and Δbehavioral uncertainty were negatively correlated across sessions. Regardless, OD activity in PMd was still positively correlated with Δbehavioral uncertainty.

Across all sessions, we observed results similar to the single session example. The PMd decoder nearly always performed better during low uncertainty trials than high uncertainty trials (*Figure 10B*), especially during the visual and delay periods. PMd decoding generally did improve at the time of movement, however the difference in decoder performance between low and high uncertainty conditions remained significant. T-Tests on the performance difference between low and high uncertainty revealed significantly better low-uncertainty performance in all behavioral periods

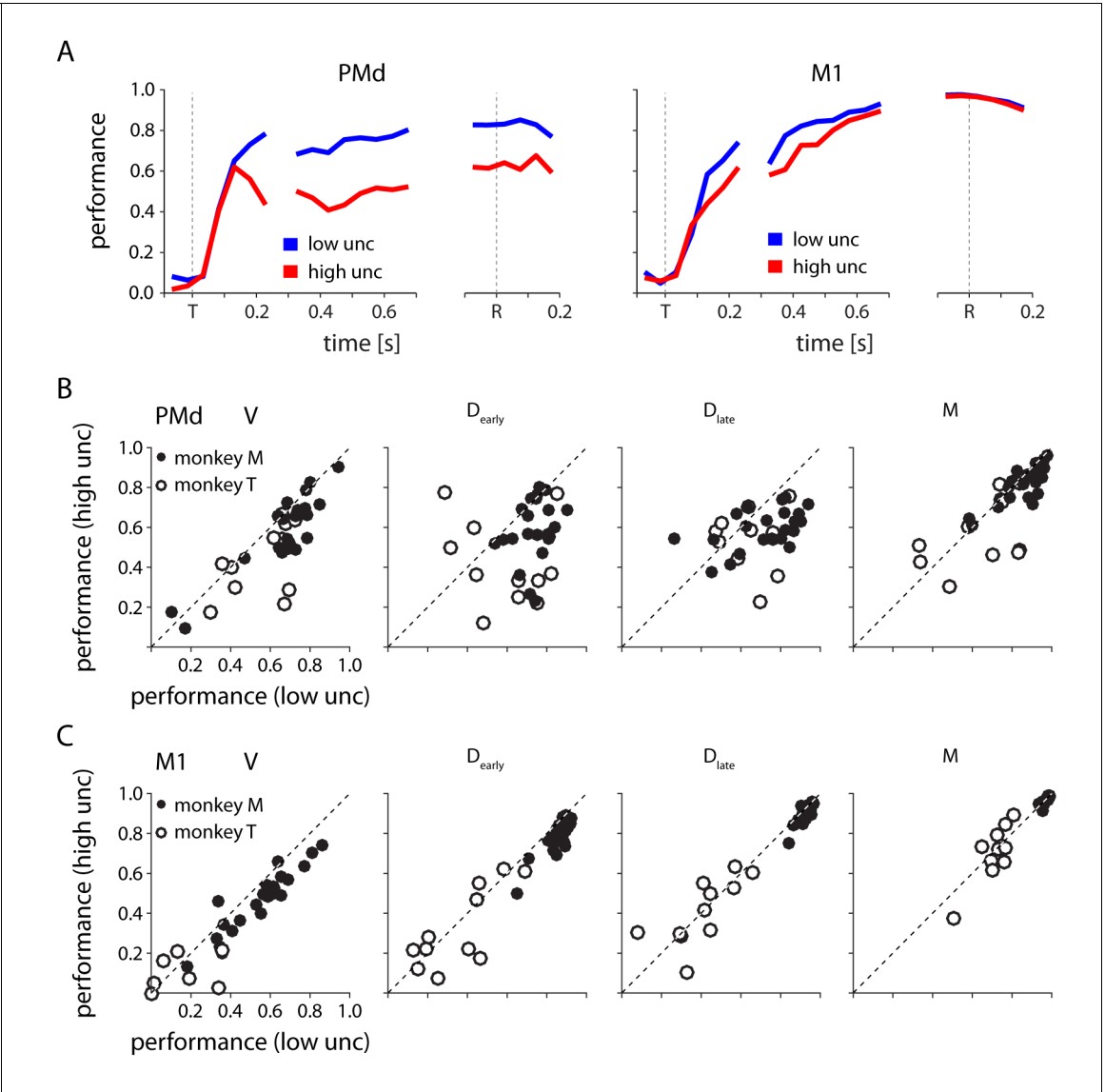

**Figure 10.** Decoding reach direction from neural activity measured on single trials from PMd and M1, based on PDs computed during center-out (zero-uncertainty) reaches. (**A**) The performance of PMd (left) and M1 (right) decoders as a function of time for one example session. Performance is defined as one minus the circular variance of the decoder error. (**B**) PMd decoder performance in low v. high uncertainty conditions for four 200 ms time windows spanning target appearance to movement in all sessions for both monkeys. Each point represents a single session from monkey M (closed) or monkey T (open) (**C**) Same as in **B**, but for M1.

for monkey M (p<0.05) and all except the movement time period for monkey T (discounting sessions with overall poor decoding, see Materials and methods). In M1, decoding performance was also slightly better for low uncertainty trials during the visual and delay periods (t-test, p<0.05), although only for monkey M (*Figure 10C*). This effect of uncertainty was much smaller than that observed in PMd. At the time of movement there was no bias in performance between low and high uncertainty trials. In general, we found decoding from M1 to be more accurate than from PMd, and less affected by uncertainty – especially at the time of movement.

## Controls

We considered several alternate explanations for the effects of uncertainty, including differences in the visual stimuli, inhomogeneous distribution of the target prior over sessions, and variations in the kinematics of reaching.

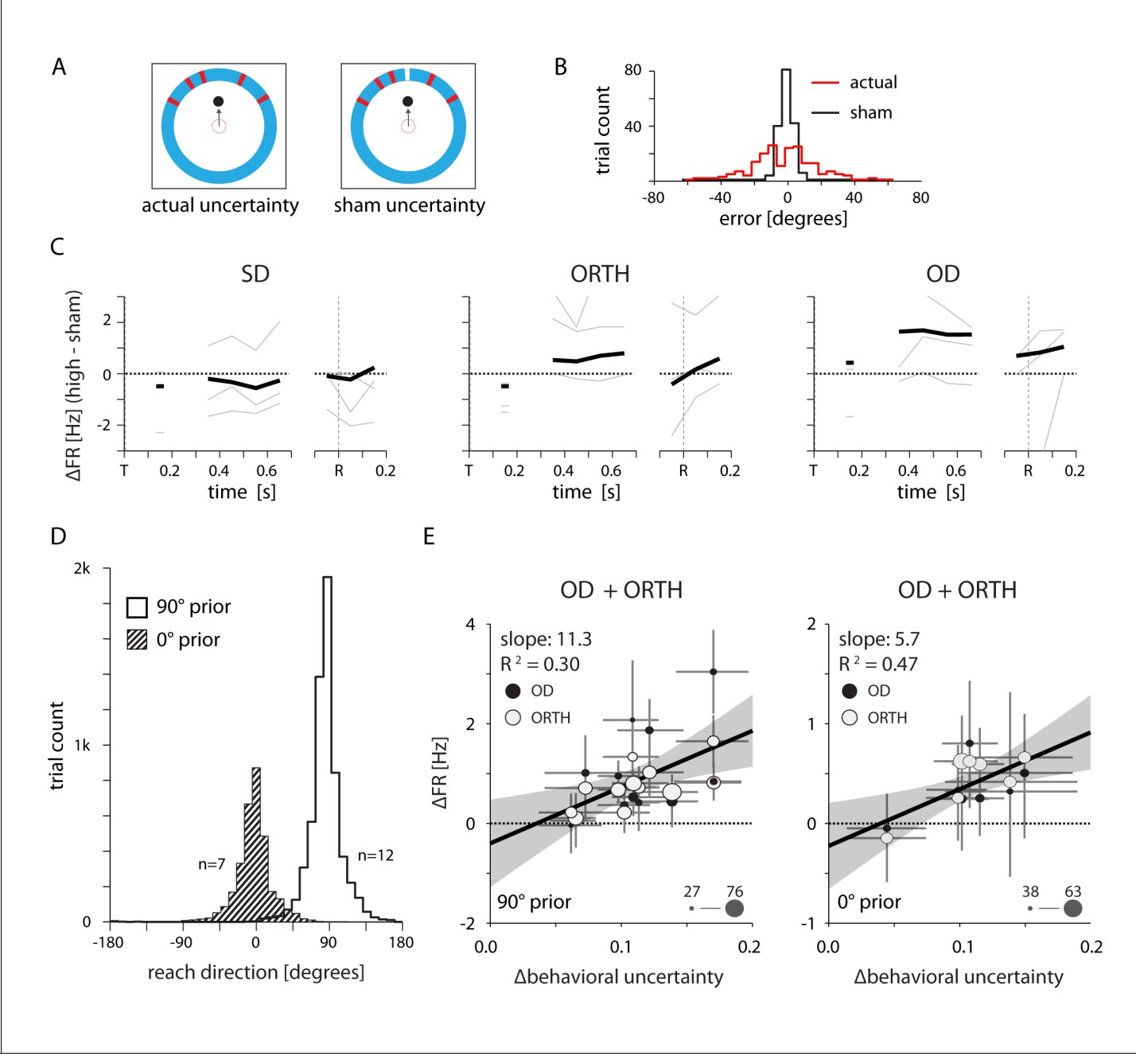

**Figure 11.** Neural effects cannot be explained by either the visual qualities of the target cue or changes in the average reach direction across sessions. (A) Design of a control experiment to test whether the uncertainty-related effect could be explained solely by differences in the visual stimuli between conditions. Half of the trials contained a high-uncertainty cue (top left) and the other half contained sham high-uncertainty trials that included an additional line of a different color to indicate the veridical target location (top right). (B) Reaching errors were much smaller for the sham trials, indicating that the monkey learned to rely on the veridical cue. (C) Thin lines indicate the average difference in firing rate between actual and sham uncertainty trials for individal sessions. Heavy lines mark the mean across sessions. OD activity was higher during actual high uncertainty trials, despite the nearly equivalent visual properties. (D) Control to test whether the neural effects could be explained by differences in the average target location across sessions. We selected two groups of sessions that each contained a consistent average reach direction. (E) Correlations between changes in OD and ORTH activity and Δbehavioral uncertainty for the two groups of sessions, 500–700 ms after target appearance. OD and ORTH activity within each group of sessions still correlated with Δbehavioral uncertainty.

The following figure supplement is available for figure 11:

**Figure supplement 1.** Kinematic controls.

To test for possible visual effects, we performed three control sessions with a single monkey (monkey M) in which half of the high-uncertainty trials contained an additional, different colored line segment at the correct target location (*Figure 11A*). These sham trials had almost exactly the same visual properties as high uncertainty trials, but did not actually induce any uncertainty. The monkey learned to rely entirely on the new cue line (*Figure 11B*). Comparing the difference in activity between actual high uncertainty and sham uncertainty trials, we found that OD (and to some extent ORTH) activity was greater only for the actual high uncertainty condition (*Figure 11C*). This suggests that our main finding of uncertainty-related changes in ORTH and OD activity cannot be explained simply as the result of differences in the visual information.

We also considered the possibility that the effects on neural activity resulted from changes in the average target location (and subsequently the average reach direction) across sessions. We tested this possible explanation by separately analyzing groups of sessions with a shared average target direction. *Figure 11D* shows the distribution of reach directions for two groups of sessions for Monkey M in which the average target location was at either 0 or 90 degrees. Analyzing these two sets of sessions separately revealed a positive correlation between changes in OD/ORTH activity and Δbehavioral uncertainty (*Figure 11E*) that was very similar to the full data set (*Figures 7* and *8*).

We anticipated that both the reaction time and peak speed might be affected by the target uncertainty, and might indirectly give rise to the firing rate changes we observed in PMd. In fact, these differences were rather small, but to test this possibility, we resampled the trials in each session to reverse the sign of the uncertainty effect on either reaction time or peak speed (*Figure 11— figure supplement 1*). These manipulations had no effect on the correlation between PMd activity and Δbehavioral uncertainty, indicating that the difference was not simply driven by kinematics.

## Discussion

### Summary

In this study, we set out to examine the neural effects of uncertainty on the motor system during a target estimation task. We showed that when visual cues of target location were made less informative, monkeys biased their reach direction toward the average target location that they had learned over the course of previous trials (their *prior* estimate) in a Bayesian-like manner. Activity in dorsal premotor cortex (PMd) changed systematically as a function of the resulting uncertainty in the monkeys' final estimate of target location, with higher uncertainty leading to higher activity in PMd neurons. This effect was not present in primary motor cortex (M1). The extent to which uncertainty affected the activity of PMd neurons depended on their directional tuning properties. Neurons with preferred directions aligned to the ultimate reach direction showed no correlation with uncertainty, while those with orthogonal or opposite direction tuning displayed significant increases in activity with increased uncertainty. This can be interpreted as an increase in uncertainty causing in increase in the representation of less likely movements directions.

### Representation of the process of target selection versus estimation

The uncertainty-related effect in PMd was present not only during movement planning, but also during execution – a result not readily predicted from previous studies. Several studies have recorded from PMd neurons as monkeys chose between multiple potential reach options (*Cisek and Kalaska, 2005*; *Coallier et al., 2015*; *Klaes et al., 2011*; *Pastor-Bernier and Cisek, 2011*; *Thura and Cisek, 2014*). Some even included ambiguous cues (*Coallier et al., 2015*; *Thura and Cisek, 2014*), which we might expect to induce uncertainty in the monkeys' decisions. The resulting representations of potential actions in PMd did, in some sense, reflect the monkey's uncertainty in the choice prior to movement execution. However, in no studies before ours did the activity changes induced by an ambiguous cue persist throughout movement execution. One study that used gradually accumulating evidence to trigger movement choice (*Thura and Cisek, 2014*) found that prior to movement, greater ambiguity in the cue resulted in a stronger representation of the target that was ultimately not selected. They observed no effect on activity corresponding to the selected target, which reached a consistent peak about 300 ms prior to movement initiation. These observations are well in line with our own results. However, at the time of movement initiation they found no ambiguity-related effects on activity, for either the neurons tuned to the selected target or the non-selected

target. This is at odds with our finding of a persistent effect of uncertainty on the representation in PMd throughout the execution of movement.

That we did not observe a resolution in the reach representation prior to movement execution may reflect a difference in the decision-making processes associated with target *estimation* and target *selection*. Inherent to target selection is the knowledge that the correct action will only be one of several mutually exclusive options. This constraint represents additional task-relevant information that can (and should) be integrated into the decision-making processes within sensorimotor areas like PMd. In a target selection task, reaching anywhere that is not an explicit target will lead to failure. It is therefore reasonable for the system to enforce a policy that before initiating a reach, the representation must only reflect one of the explicit target options. However, in target estimation tasks there are no such constraints on the executed action, allowing for a broader movement representation.

## Differences in the roles of PMd and M1

The different neural responses observed during target selection and target estimation has important implications for the assumed roles of M1 and PMd. Results from target selection tasks suggest that movement decisions are made within PMd as the result of a biased neural competition between potential actions (*Cisek, 2007*; *Gallivan et al., 2016*; *Pastor-Bernier and Cisek, 2011*). This interpretation – that PMd ultimately decides on the action – is especially convincing given the previous observations that decision-related variables (e.g., cue ambiguity, uncertainty, etc.) had no effect on movement representations in PMd at the time of movement initiation (*Thura and Cisek, 2014*). However, we observed uncertain representations that persisted throughout movement execution, which indicates that PMd may not necessarily be the final step in the motor decision-making process.

Our results suggest that PMd does not actually decide which movement to execute, but rather that it maintains a continuously updated estimate of the distribution of potentially useful actions. This distribution is likely dependent on a number of factors, including current sensory information and prior experience, as well as constraints specific to the task (e.g., selecting between mutually exclusive targets). In our study, the distribution of actions represented in PMd changed according to the monkey's behavioral uncertainty, but was relatively static within a single trial. At no point did the distribution resolve into a single unambiguous reach representation. M1, on the other hand, seemed relatively unaffected by uncertainty and consisently reflected the direction of the executed reach. These findings imply that PMd is not solely responsible for 'deciding' which movement to execute, but instead contains only a noisy representation of potential reach directions that must be interpreted in some way by downstream areas like M1. Thus, we suggest that the processing that occurs in the connections between PMd and M1 'denoises' the PMd representation to provide a single, unambiguous movement decision.

The reach decoding results (*Figure 10*) support the interpretation that action decisions do not arise solely from PMd. Decoding performance at the time of movement initiation was significantly higher compared to delay-period levels, especially for high uncertainty trials. However, we still observed a consistent bias towards better decoding performance on low uncertainty trials. M1, on the other hand, showed a steady increase in reach-related information leading up to movement initiation that was slightly skewed towards better accuracy under low uncertainty. At the time of movement initiation, M1 was able to decode reach direction with high accuracy, regardless of uncertainty condition. From these observations, we speculate that the decision about where to reach is not explicitly determined in PMd, but rather in the connections between PMd and M1. The noisy representation of potential actions in PMd appears to be interpreted in some way by M1, ultimately producing a single unambiguous motor command.

The process by which M1 obtains a final movement representation could potentially occur through a *maximum a posteriori* (MAP) readout of the PMd representation. This kind of mechanism is not only consistent with the results of the current and previous studies, but could potentially explain the neural basis of sensorimotor learning. For example, we would expect that in very high uncertainty conditions (e.g., a novel behavioral task), PMd might retain nearly equal representations of all possible movements. As a consequence, small fluctuations due to noise within PMd would cause large variability in a downstream readout, driving exploration of the environment. As learning

progressed and uncertainty decreased, the distribution in PMd would narrow and motor output would begin to converge on the optimal movement decision.

## PMd reflects uncertainty in the decision, not the visual cue

Our task varied the monkeys' uncertainty in target estimation by manipulating both the history of target distribution and the noise in visual cues. We found that PMd activity changed not as a function of the weighting of either of those two pieces of information, but rather in proportion to the total uncertainty in the final decision. Thus PMd contains uncertainty-related information pertaining to the final action, which encompasses more than just the reliability of the visual cue. Additionally, if uncertainty in visual information were the sole driving force of changes in PMd planning- and execution-related activity, we would have observed very little difference in activity across sessions, since the visual cue properties were largely equivalent for all sessions. Instead, we found that activity modulated with the total behavioral uncertainty, which is a combination of visual uncertainty and prior expectation. This suggests that PMd likely reflects the combined uncertainty of all information sources relevant to a movement decision.

## Comparison with existing theoretical models of uncertainty

There exist a number of theoretical models that address the potential neural representation of uncertainty (*Deneve, 2008*; *Hinton and Sejnowski, 1983*; *Hoyer and Hyvarinen, 2003*; *Ma et al., 2006*; *Zemel et al., 1998*). The predictions from these models encompass a wide range of neural behaviors, including temporal dynamics (*Deneve, 2008*) and variability in spike timing (*Deneve, 2008*; *Hoyer and Hyvarinen, 2003*). Unfortunately, our experimental design prevents us from performing fair and comprehensive tests of these model predictions. For example, our use of a static visual cue and instructed delay limits the potential interpretations regarding dynamic uncertainty codes. For these reasons, we hesitate to make any strong statements about the validity of any given model.

Despite the limitations of our experimental design, our results do bear some resemblance to admittedly simplistic interpretations of a few theoretical models. A probabilistic population code (PPC) model predicts that firing rates across a population should reflect the probability distribution – high uncertainty should therefore result in lower peak activity and higher non-peak activity (*Ma et al., 2006*). We did indeed observe an increase in non-peak activity with increased uncertainty, and the spatiotemporal activity plots in *Figure 5* do convincingly resemble probability distributions of reach direction. However, we did not see any consistent decrease in the peak activity with increasing uncertainty, which prevents us from interpreting the population activity as representing a true probability distribution. Our findings also argue against the concept of divisive normalization, in which the total activity remains equivalent when representing multiple potential targets (*Cisek and Kalaska, 2005*; *Pastor-Bernier and Cisek, 2011*), at least in the context of target estimation.

## Conclusions

Our results provide new insight into the behavior of PMd during movement planning. It is already well established that PMd can simultaneously represent all potential actions when faced with multiple, mutually exclusive visual targets (*Bastian et al., 2003*; *Cisek and Kalaska, 2005*). Our results provide the additional observation that PMd also represents and retains a distribution of potential motor plans that are not explicitly presented, but arise as possibilities during uncertain target estimation. The question of why this representation is maintained for the problem of target estimation but not target selection is an interesting one. One possibility is that it is simply an unavoidable result of noisy inputs to PMd. That is, in the absence of explicit reach targets, the fidelity of the representation in PMd may be limited by the quality of available information. On the other hand, maintaining heightened representations of alternative movements in high uncertainty conditions may be useful to the sensorimotor system for more rapid error correction or to drive subsequent motor learning. Experiments designed to test these alternatives could help to further our understanding of the role of PMd in movement planning.

## Materials and methods

All surgical and experimental procedures were fully consistent with the guide for the care and use of laboratory animals and approved by the institutional animal care and use committee of Northwestern University under protocol #IS00000367.

### Behavioral task

The monkeys were seated in front of a vertical monitor and controlled an on-screen cursor using a planar robotic manipulandum. The behavioral task involved two or more blocks of trials. In the first block, monkeys performed a basic center-out reaching task with an instructed delay period. The monkey held the cursor within a central target for a random length center-hold period (700–1000 ms), after which a target (15 degrees wide) appeared in one of eight well-defined locations, distributed equally around an outer ring (*Figure 1A*, top). Following an additional random delay period (700–1000 ms) the center target disappeared and the monkey received an auditory signal cueing him to reach to the outer target. Upon reaching the outer ring, the cursor froze. If the cursor was within the target, the monkey heard a success tone and received a small amount of juice. Otherwise, the monkey heard a failure tone and received no juice reward.

In the remaining (uncertainty) trial blocks, the target locations $\theta$ were not distributed uniformly among eight locations as before, but were instead selected randomly from a von Mises (circular normal) *prior*

$$f(\theta) = \frac{e^{\kappa \, \cos(\theta - \mu)}}{2\pi \, I_0(\kappa)} \tag{1}$$

The mean of this prior distribution (μ) was always fixed for the duration of a session, but could vary in width (κ) across trial blocks. Additionally, during uncertainty trials the monkeys did not receive veridical visual cues about the target until the end of the trial. Instead, during planning and execution they were only shown several small lines (five for monkey M, ten for monkey T) sampled from a *likelihood* distribution (also von Mises) centered on the target location (*Figure 1A*, bottom). These lines gave the monkey information about the target location, but with different levels of uncertainty depending on the variance of the distribution. Each session contained two different likelihood distributions, which were randomly interspersed across trials. The exact parameters used for each session are provided in *Figure 1—source data 1*. Upon reaching to the outer ring, the cursor froze and the ambiguous cue lines were replaced with the actual target (15 degrees, all conditions). The monkey subsequently received (or did not receive) reward as in the center-out trial block.

Although we directly specified the variance (and therefore uncertainty) in the target distribution and the visual cue, the monkeys' subjective estimates of those parameters could deviate considerably from their true values. We therefore used the monkeys' actual responses throughout the session to estimate two values: the monkeys' weighting of the current visual cue, and the total uncertainty remaining in the monkeys' final estimate of the required reach direction. To do this, we assumed a Bayesian-like model of cue integration in which the final estimate was the product of likelihood (visual cue) and prior (distribution of target locations) probability distributions. We modeled both of these as von Mises distributions. The product of two von Mises distributions can be approximated by a third, with mean

$$\mu_3 = \mu_1 + tan^{-1}\left(\frac{\sin(\mu_2 - \mu_1)}{\frac{k_1}{k_2} + \cos(\mu_2 - \mu_1)}\right) \tag{2}$$

To obtain an estimate of the relative weighting of the visual cue for each uncertainty condition, we substituted the true target centroid location for $\mu_2$, the true average target location for $\mu_1$, and then fit $\left(\frac{k_1}{k_2}\right)$ to minimize the sum of the squared residuals between the model outputs and the monkeys' actual reach directions. The resulting equation for $\mu_3$ describes the general function relating the centroid of the visual cue and the reach direction. (red and blue lines; *Figure 1B*). Except for cases in which $|\mu_2 - \mu_1|$ is very large, this can be suitably approximated by the linear function

$$\mu_3 \cong \mu_1 + \frac{k_2}{k_1 + k_2} (\mu_2 - \mu_1) \tag{3}$$

In all further analysis, we use the slope term $\left(\frac{k_2}{k_1+k_2}\right)$ as a proxy for our estimate of the monkeys' relative weighting of the visual cue with respect to the summed prior and likelihood uncertainty. Slopes close to one represent high reliance on the visual cue, while slopes close to zero represent high reliance on the average prior target location.

The slope metric described above reveals only the monkeys' *relative* uncertainties in the likelihood and prior. It does not contain any information about the total magnitude of uncertainty present in the monkeys' decisions. We estimated this total uncertainty from the monkeys' behavior, by calculating the angular dispersion of the residuals from each behavioral fit like those shown in *Figure 1B*. It is important to note that the behavioral uncertainty can be affected by uncertainty in the estimate of average target location, uncertainty in the visual cue, and potentially other internal variables affecting the monkeys' behavior that we did not control (e.g., motivation, attention).

## Neural recordings and analysis

Throughout the experiments we recorded from neurons in M1 and PMd (*Figure 2A*) using chronically implanted 96-channel microelectrode arrays (Blackrock Microsystems, Salt Lake City UT). We identified single neurons from each session using offline sorter by isolating clusters within a principle component space projected from the waveform shapes of putative neurons (Plexon Inc., Dallas TX). There was likely significant overlap between sessions of the populations of recorded neurons, but we made no effort to track the identity of neurons across sessions. On each session, we used the activity from the center-out block of trials (zero uncertainty, eight target locations) to characterize the directional tuning characteristics of all neurons. Since many neurons (especially those in PMd) can have complex temporal profiles, we calculated preferred directions (PDs) in three distinct time periods: *visual* (50–250 ms after target appearance), *delay* (300–700 ms post-target), and *movement* (0–200 ms after initiation of the reach) using a generalized linear model with Poisson noise:

$$\lambda = \exp[\alpha + \beta\cos(\theta - \theta^*)] \tag{4}$$

where $\lambda$ is a vector of firing rates across trials, $\theta$ is a vector of reach directions, $\theta^*$ is the preferred direction, and $\alpha$, $\beta$ are scaling parameters. For each neuron, we also obtained confidence bounds on the fit parameters through bootstrapping. A neuron was only considered to be significantly tuned if 95% of the bootstrapped estimates of $\theta^*$ were within forty-five degrees of the mean estimate. Due to the lower neuron count for monkey T, we relaxed this constraint to accept neurons with bootstrapped PDs within ninety degrees of the mean. For all analyses, we used only the preferred directions calculated within the appropriate time period (for example, delay-period tuning for all delay-period analyses). When analyzing a given time period, we excluded neurons without significant tuning in that period. Full details on the numbers of tuned neurons for all sessions is provided in *Figure 1—source data 1*.

## Single trial decoding analysis

We used a simple decoding approach based on each neuron's PD, computed from data collected during the center-out (zero-uncertainty) task. We first divided neurons according to their PDs, creating sixteen bins of 22.5 degrees each. We then averaged the activity of all neurons within each bin (after first subtracting pre-target baseline activity levels) and fit a cosine to the resulting activity profile. The peak of this cosine defined the decoded reach direction. We characterized decoder performance for each uncertainty condition as one minus the circular variance of decoder error. Circular variance is bounded by 0 and 1. Therefore, a performance of 0 represents that the decoder did no better than random guessing, and a performance of 1 represents perfect decoding of the reach direction. This metric is similar to VAF, except it is not normalized by the total variance of the reach distribution. This is important for our dataset, which contained very non-uniform distributions of reach directions. It provides a fair comparison of decoder performance regardless of differences in distributions between sessions or uncertainty conditions.

We assessed the effect of uncertainty condition on decoding performance by performing t-tests on the distributions of differences between low and high uncertainty conditions for each monkey and time period. For monkey T, low neuron counts made decoding on a trial-by-trial basis much less accurate. Therefore, when assessing biases, we only included sessions in which the decoder performance on low uncertainty trials was greater than 0.5.

## Additional information

### Funding

| Funder | Grant reference number | Author |
|---|---|---|
| National Institute of Neurological Disorders and Stroke | R01 NS074044 | Konrad P Kording<br>Lee E Miller |

The funders had no role in study design, data collection and interpretation, or the decision to submit the work for publication.

### Author contributions

BMD, Conception and design, Acquisition of data, Analysis and interpretation of data, Drafting or revising the article; PR, KPK, LEM, Conception and design, Analysis and interpretation of data, Drafting or revising the article; PAW, Conception and design, Acquisition of data, Analysis and interpretation of data

### Author ORCIDs

Lee E Miller, http://orcid.org/0000-0001-8675-7140

### Ethics

Animal experimentation: All procedures were approved by the Northwestern University Institutional Animal Care and Use Committee and were consistent with the Guide for the Care and Use of Laboratory Animals. Protocol number #IS00000367.

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
