## [Decision Letter]

Thank you for submitting your article "Uncertainty leads to persistent representations of alternative movements in PMd" for consideration by *eLife*. Your article has been reviewed by two peer reviewers, and the evaluation has been overseen by Michael Frank as the Reviewing Editor and Timothy Behrens as the Senior Editor. One of the two reviewers has agreed to reveal his identity: Veit Stuphorn.

The reviewers have discussed the reviews with one another and the Reviewing Editor has drafted this decision to help you prepare a revised submission.

Summary:

This manuscript reports neuronal activity in dorsal premotor (PMd) and primary motor cortex (M1) when monkeys select arm movements with varying degrees of uncertainty about the correct movement direction. The main finding is that in PMd, but not M1, increased uncertainty leads to increased activation of neurons representing the non-chosen arm direction. This work builds on and extends the work by Cisek and coworkers in a number of important ways. First, it indicates that PMd does not represent the decision process, but rather represents the distribution of all potentially correct actions. Second, it provides important information about the way uncertainty is represented in the brain.

Essential revisions:

Both reviewers agreed that the topic is very timely, the experiment is well designed and has potentially very interesting features, and that the contribution has lots of potential. However, they also highlighted a number of issues that need to be cleared up before we can further evaluate the paper. The comments of the two reviewers are amalgamated below.

1) Fundamental information on the experimental details are lacking. How were the animals rewarded in the different prior and likelihood conditions, what was considered "correct" behavior? How did the animals perform in the different conditions (RTs, error rates, endpoint distributions, etc.? Without this crucial information on the task design and behavior it is impossible to interpret the results, e.g. the behavior in Figure 1. Were the prior and the likelihood centroids evenly distributed over all directions? In other words, which percentage of sessions was centered on the 90 degrees direction during the whole course of the training of each animal? Depending on the reward structure in the task and the performance of the animals, the current interpretation could be substantially confounded. If in the end we have to admit that the results can be explained by spatial value-based target preference (via reward expectancy) or overtrained default behavior (reaching to 90 degrees) then the relevance or novelty of the finding could be substantially hampered.

2) The different possible conceptualizations of "uncertainty" come rather late in the Introduction; the early Introduction sometimes refers to probabilistic uncertainty for selection among mutually exclusive targets (typical dual target tasks); sometimes it refers to a motor control perspective with blurred target information or just the fact that the target has to be memorized instead of being visible. More precise statements and more adequate references (see specific comments below) would be possible if the different scenarios (target estimation vs. target selection) were clearly distinguished from early on. But in their own interpretation in the remaining manuscript, the authors seem to be not clear about what concept they actually want to discuss. Repeatedly they use nomenclature suggesting that they speak of target selection (e.g. when calling the higher activity of OD neurons during high uncertainty a 'persistent representations of alternative movements') while their task is about target estimation. Moreover, the activity level of OD neurons in this case is still below baseline, which seems difficult to sell as "representation" of a potential movement. For such statement, I would need some behavioral indication that the animal ever considers an alternative movement to the one towards the 'blurred' target stimulus. (One should stay skeptical that a biased competition model is invalidated by the presented data if the task does not demand the animal to choose between competing behavioral alternatives, but rather just tries to estimate the one demanded movement most precisely – I doubt that target estimation and target selection are interchangeable cognitive demands in this respect.)

3) Relatedly, one of the most interesting points in this study is the information we might be able to gain from it regarding the representation of uncertainty. However, in this study the authors just state in the Discussion section that the data do not 'fully explain any of the theories', but are best explained by a probabilistic population code model. This is clearly not sufficient. The author should address this issue more directly and at least report the results of their attempts to test the various uncertainty models. Otherwise, the statement is simply not backed up by evidence and a major source of interest in the manuscript is missing.

4) It did not become clear to me how the results and conclusions precisely relate to the earlier cited study by Bastian et al. (2003). Bastian et al. 2003 recorded in M1, not PMd, but otherwise, didn't they show a collapse in the peaks of the population tuning onto the center of multiple (2 or 3) potential reach targets; different to how the earlier results are discussed here, it seems to me that this earlier finding is not evidence for a simultaneous co-representation of 3 targets in parallel, but could mark an "average" target representation or the centroid of a prior or likelihood distribution, similar to here? The manuscript would benefit from a more detailed discussion of earlier findings.

5) Also lacking is information on the way neurons were counted over sessions. Since chronic arrays were used for recording, more information is needed on when a channel was counted as "neuron" (I presume spike sorting was applied; does the N in table in supplement also include "multi-unit" channels?), and – more importantly – how the same channel was used towards the statistics across different sessions.

6) The figures are notoriously lacking information on the N number of data samples going into the analysis, the nature of the error bars/confidence limits shown, the meaning of "delta" in the several y-axis showing delta-FR, and whether we look at data from one animal or both animals. This problem culminates in Figure 6—figure supplement 1 showing results for the animal T which do not match (SD conditions) the results of the animal shown in the main manuscript (until then I was unclear about the fact that I looked at data from only one animal in the main figure). But how is the discrepancy between animals resolved, how does it affect conclusions? Is Figure 5 an average of both animals? Why were there only 3 sessions recorded in monkey T despite visible differences in behavior to the other monkey (Figure 1)?

7) The off-target activity increase with target uncertainty seems not to be motivated apriori by Bayes models of uncertainty processing; actually, it remains elusive how well the results fit to corresponding model predictions, e.g. on how prior and likelihood distributions should be integrated and how each should affect behavior/neural tuning. The finding of OD activity increase seems rather unspecific and less conclusive than it could be with this experimental design and theoretical framework with which the study is motivated.

8) The authors find that PMd neurons that encode the non-chosen direction increase their activity under increased uncertainty, while the PMd neurons that encode the chosen direction show unchanged activity. This activity pattern is interesting, because it argues against divisive normalization as has been postulated by a number of authors, including Cisek (Pastor-Benier & Cisek, 2011). Specifically, the increased activity in the non-chosen action representing neurons should have diminished the activity in chosen action representing ones. The authors should discuss this point.

[Editors’ note: what now follows is the decision letter after the authors submitted for further consideration.]

Thank you for resubmitting your work entitled "Uncertainty leads to persistent effects on reach representations in dorsal premotor cortex" for further consideration at *eLife*. Your revised article has been favorably evaluated by Timothy Behrens (Senior editor), Michael Frank (Reviewing editor), and two reviewers, one of whom is Veit Stuphorn.

Both reviewers agreed that the manuscript has been substantially. The major issue is a request by Reviewer 1 to do a decoding analysis, the results of which will be informative about the interpretation offered (even if that interpretation is tentative and not completely essential).

Reviewer #1:

The authors have done a very good job in clarifying the main conceptual issues in the paper. I have one remaining question. I would agree with the authors that their findings indicate that PMd might not be the final step in skeletomotor decision-making. However, it would be useful to see a decoding analysis. Taking activity recorded in PMd and M1, how well can the ultimate arm movement direction be decoded at the various times in the trial in the different trial periods. If the hypothesis of the authors is correct, one would expect a strong and consistent encoding of chosen direction in M1 (potentially also earlier than in PMd). On the other hand, PMd should not or only very weakly encode chosen action, in particular in the high-uncertainty condition. Given the data in Figure 5, I am confident that the M1 prediction will turn out to be true, but I am not sure about PMd. How would the hypothesis of the authors be affected, if PMd would also encode chosen direction? Could PMd and M1 just be links in a set of cascading networks in which the initial selection simply gets sharpened?

Reviewer #2:

The authors did an excellent job in revising the manuscript. They re-wrote substantial parts of the manuscript to address my main request for a better distinction between uncertain selection among well specified targets and target uncertainty, thereby making this topic a main theme of the paper which is consistently and very nicely dealt with throughout.

---

## [Author Response]

*Essential revisions:*

*Both reviewers agreed that the topic is very timely, the experiment is well designed and has potentially very interesting features, and that the contribution has lots of potential. However, they also highlighted a number of issues that need to be cleared up before we can further evaluate the paper. The comments of the two reviewers are amalgamated below.*

1) Fundamental information on the experimental details are lacking. How were the animals rewarded in the different prior and likelihood conditions, what was considered "correct" behavior? How did the animals perform in the different conditions (RTs, error rates, endpoint distributions, etc.? Without this crucial information on the task design and behavior it is impossible to interpret the results, e.g. the behavior in Figure 1. Were the prior and the likelihood centroids evenly distributed over all directions? In other words, which percentage of sessions was centered on the 90 degrees direction during the whole course of the training of each animal? Depending on the reward structure in the task and the performance of the animals, the current interpretation could be substantially confounded. If in the end we have to admit that the results can be explained by spatial value-based target preference (via reward expectancy) or overtrained default behavior (reaching to 90 degrees) then the relevance or novelty of the finding could be substantially hampered.

We have added several new figures as well as explanatory text that enlarges on the details of the task design and the monkeys’ performance. We have carefully considered those aspects of the behavior that might have confounded the results. We address the concerns that neural effects may have resulted from reaction time (Figure 11—figure supplement 1), reach speed (Figure 11—figure supplement 1), or the distribution of average target locations (Figure 9). We also include extensive information about target distributions and other behavior-related parameters in [Supplementary-material SD1-data].

2) The different possible conceptualizations of "uncertainty" come rather late in the Introduction; the early Introduction sometimes refers to probabilistic uncertainty for selection among mutually exclusive targets (typical dual target tasks); sometimes it refers to a motor control perspective with blurred target information or just the fact that the target has to be memorized instead of being visible. More precise statements and more adequate references (see specific comments below) would be possible if the different scenarios (target estimation vs. target selection) were clearly distinguished from early on. But in their own interpretation in the remaining manuscript, the authors seem to be not clear about what concept they actually want to discuss. Repeatedly they use nomenclature suggesting that they speak of target selection (e.g. when calling the higher activity of OD neurons during high uncertainty a 'persistent representations of alternative movements') while their task is about target estimation. Moreover, the activity level of OD neurons in this case is still below baseline, which seems difficult to sell as "representation" of a potential movement. For such statement, I would need some behavioral indication that the animal ever considers an alternative movement to the one towards the 'blurred' target stimulus. (One should stay skeptical that a biased competition model is invalidated by the presented data if the task does not demand the animal to choose between competing behavioral alternatives, but rather just tries to estimate the one demanded movement most precisely – I doubt that target estimation and target selection are interchangeable cognitive demands in this respect.)

We have extensively rewritten both the Introduction and Discussion to deal with these important points.

1) We now establish the difference between uncertain target selection vs. estimation, and make our discussion of these points much more consistent.

2) We have more clearly discussed the implications of an increase in OD neuron activity. A single movement representation results in a continuum of neural responses across neurons – typically represented as a cosine-like function. Due to the continuous nature of this response across the population, we see little reason to believe that sub- and super-baseline activity differ fundamentally in their contributions to overall movement representation. Rather the spatial activity pattern across PMd simply reflects the current probability distribution of potential movement directions. Thus, for clarity’s sake we decided to use consistent nomenclature for cross-condition comparisons of activity. We therefore refer to increases in activity as increases in representation, regardless of the final relationship to baseline.

3) We have expanded the discussion of how our task may or may not support a competition model hypothesis and acknowledge that target selection and target estimation are not equivalent tasks. In fact, we suggest that our results can help improve modeling efforts (including the competition model) by adding target estimation to the large base of target selection literature.

3) Relatedly, one of the most interesting points in this study is the information we might be able to gain from it regarding the representation of uncertainty. However, in this study the authors just state in the Discussion section that the data do not 'fully explain any of the theories', but are best explained by a probabilistic population code model. This is clearly not sufficient. The author should address this issue more directly and at least report the results of their attempts to test the various uncertainty models. Otherwise, the statement is simply not backed up by evidence and a major source of interest in the manuscript is missing.

An earlier version of this manuscript (submitted elsewhere) had a more extensive discussion of the relation between our results and those predicted by several prominent models. This proved to be a frustration both for us and reviewers, in part because of the imprecision of the predictions made by some of those models, and by their dependence on particular experimental conditions that were not always well met by our study. This led us to back away from stronger assertions about particular models in favor of more general comments. We have attempted to provide a bit more detail on the relation between particular models and our experimental results but we also feel that the better description of the relation between our results and the previous target selection studies in PMd have taken this paper in a somewhat different direction than that of a general discussion of uncertainty representation.

4) It did not become clear to me how the results and conclusions precisely relate to the earlier cited study by Bastian et al. (2003). Bastian et al. 2003 recorded in M1, not PMd, but otherwise, didn't they show a collapse in the peaks of the population tuning onto the center of multiple (2 or 3) potential reach targets; different to how the earlier results are discussed here, it seems to me that this earlier finding is not evidence for a simultaneous co-representation of 3 targets in parallel, but could mark an "average" target representation or the centroid of a prior or likelihood distribution, similar to here? The manuscript would benefit from a more detailed discussion of earlier findings.

The PMd-related references to the Bastian study were incorrect. We have removed them and made extensive changes to all sections referencing this and other existing literature.

5) Also lacking is information on the way neurons were counted over sessions. Since chronic arrays were used for recording, more information is needed on when a channel was counted as "neuron" (I presume spike sorting was applied; does the N in table in supplement also include "multi-unit" channels?), and – more importantly – how the same channel was used towards the statistics across different sessions.

This was a significant oversight. We did, indeed, identify single neurons using the industry standard offline sorter, and have now more carefully reported the numbers of neurons recorded in each session in [Supplementary-material SD1-data]. In so doing, we recognize that each session will include a varied mixture of new and previously recorded neurons. However, because of the difficulty in establishing a neuron’s identity with high confidence, we have treated analyses as though each recorded ensemble is representative of the full population. This lack of independence will have the greatest potential effect on Figure 6, Figure 7, and Figure 8, for which we compute statistics across sessions. However, the slopes and R^2^ computed in Figure 6 and Figure 7 use only the session averages, and do not rely on combining units across days.

6) The figures are notoriously lacking information on the N number of data samples going into the analysis, the nature of the error bars/confidence limits shown, the meaning of "delta" in the several y-axis showing delta-FR, and whether we look at data from one animal or both animals. This problem culminates in Figure 6—figure supplement 1 showing results for the animal T which do not match (SD conditions) the results of the animal shown in the main manuscript (until then I was unclear about the fact that I looked at data from only one animal in the main figure). But how is the discrepancy between animals resolved, how does it affect conclusions? Is Figure 5 an average of both animals? Why were there only 3 sessions recorded in monkey T despite visible differences in behavior to the other monkey (Figure 1)?

We have added Information about numbers of neurons to Figure 6, Figure 7 and Figure 8, specified confidence limits as necessary and improved the definition of a number of terms used throughout the paper. We also clearly identified the animal used for each figure. We have only 5 sessions for Monkey T as the array subsequently failed. However, by subsampling some of these sessions we were able to perform the same analyses as for monkey M and have now provided that as Figure 7, with a description of methodology in Figure 7—figure supplement 1.

7) The off-target activity increase with target uncertainty seems not to be motivated apriori by Bayes models of uncertainty processing; actually, it remains elusive how well the results fit to corresponding model predictions, e.g. on how prior and likelihood distributions should be integrated and how each should affect behavior/neural tuning. The finding of OD activity increase seems rather unspecific and less conclusive than it could be with this experimental design and theoretical framework with which the study is motivated.

As noted above (point 3), we have rather extensively reworked the Introduction and Discussion sections to deal with this and several other related issues more clearly and consistently.

*8) The authors find that PMd neurons that encode the non-chosen direction increase their activity under increased uncertainty, while the PMd neurons that encode the chosen direction show unchanged activity. This activity pattern is interesting, because it argues against divisive normalization as has been postulated by a number of authors, including Cisek (Pastor-Benier & Cisek, 2011). Specifically, the increased activity in the non-chosen action representing neurons should have diminished the activity in chosen action representing ones. The authors should discuss this point.*

See comments above. We appreciate that the reviewers pointed out these important points, which we believe has allowed us to improve the clarity and impact of the paper considerably. We have included comments about the implications for divisive normalization.

[Editors’ note: what now follows is the decision letter after the authors submitted for further consideration.]

*Both reviewers agreed that the manuscript has been substantially. The major issue is a request by Reviewer 1 to do a decoding analysis, the results of which will be informative about the interpretation offered (even if that interpretation is tentative and not completely essential).*

*Reviewer #1:*

*The authors have done a very good job in clarifying the main conceptual issues in the paper. I have one remaining question. I would agree with the authors that their findings indicate that PMd might not be the final step in skeletomotor decision-making. However, it would be useful to see a decoding analysis. Taking activity recorded in PMd and M1, how well can the ultimate arm movement direction be decoded at the various times in the trial in the different trial periods. If the hypothesis of the authors is correct, one would expect a strong and consistent encoding of chosen direction in M1 (potentially also earlier than in PMd). On the other hand, PMd should not or only very weakly encode chosen action, in particular in the high-uncertainty condition. Given the data in Figure 5, I am confident that the M1 prediction will turn out to be true, but I am not sure about PMd. How would the hypothesis of the authors be affected, if PMd would also encode chosen direction? Could PMd and M1 just be links in a set of cascading networks in which the initial selection simply gets sharpened?*

It had not occurred to us to try to decode reach direction, but it is an interesting addition to the story. We anticipated that M1 would decode better than PMd, and that low uncertainty would be better than high. However, we would not necessarily have predicted that M1 would also show a small delay period difference in performance, or the time course of the decoding accuracy. We have added a figure and additional discussion points.

In assessing decoder performance, we opted not to use the standard variance accounted for metric (VAF) due to non-uniform and significantly different reach distributions across sessions and between uncertainty conditions. VAF normalizes performance with respect to the total variance in the signal (in this case reach direction). This variance could differ quite dramatically between uncertainty conditions, making direct comparisons of performance problematic. Instead, we simply computed performance with respect to the maximum possible circular variance (which always has an upper limit of 1). Subtracting this value from 1 created a useful and intuitive scale between 0 (decoder chooses random directions) and 1 (decoder perfectly decodes reach directions). This metric is unaffected by the distribution of reaches and thus allowed us to compare decoder performance directly across uncertainty conditions.